# `MOCA` 🫕: Self-supervised Representation Learning by Predicting Masked Online Codebook Assignments

**Spyros Gidaris**[1], **Andrei Bursuc**[1], **Oriane Siméoni**[1], **Antonin Vobecky**[1,2,3]
**Nikos Komodakis**[4,5,6], **Matthieu Cord**[1], **Patrick Pérez**[1]

[1] *Valeo.ai*
[2] *Czech Institute of Informatics, Robotics and Cybernetics at the Czech Technical University in Prague*
[3] *Czech Technical University in Prague,Faculty of Electrical Engineering*
[4] *University of Crete*
[5] *IACM-Forth*
[6] *Archimedes/Athena RC*
*Correspondance: spyros.gidaris@valeo.com*

**Reviewed on OpenReview:** *https://openreview.net/forum?id=0dDsCaacZ0*

## Abstract

Self-supervised learning can be used for mitigating the greedy needs of Vision Transformer networks for very large fully-annotated datasets. Different classes of self-supervised learning offer representations with either good contextual reasoning properties, e.g., using masked image modeling strategies, or invariance to image perturbations, e.g., with contrastive methods. In this work, we propose a single-stage and standalone method, `MOCA`, which unifies both desired properties using novel mask-and-predict objectives defined with high-level features (instead of pixel-level details). Moreover, we show how to effectively employ both learning paradigms in a synergistic and computation-efficient way. Doing so, we achieve new state-of-the-art results on low-shot settings and strong experimental results in various evaluation protocols with a training that is at least 3 times faster than prior methods. We provide the implementation code at https://github.com/valeoai/MOCA.

## 1 Introduction

Self-supervised representation learning for Vision Transformers (ViT) (Devlin et al., 2018; Vaswani et al., 2017) has attracted a significant amount of attention over the last years (Bao et al., 2022; Li et al., 2021; He et al., 2022; Xie et al., 2022; Zhou et al., 2022; Zhai et al., 2022a; Assran et al., 2022; Peng et al., 2022). Indeed, contrary to convolutional neural networks, ViTs lack image-specific inductive bias –*e.g.*, spatial locality or weight sharing– as they treat an image simply as a sequence of tokens, each token corresponding to a local image patch. As a result, although ViTs exhibit a tremendous capacity to learn powerful image representations, they require much larger amounts of annotated data to do so. In this context, self-supervised learning aims to help transformers overcome such large training set requirements and deliver their full potential by utilizing unlabeled image data that can often be readily available even in large quantities.

When it comes to applying self-supervised learning to ViT architectures, the most prominent paradigm follows a *hide-and-predict* approach, which consists in hiding tokens on the input and training the model to predict information for the missing tokens, *e.g.*, predicting token ids (Bao et al., 2022) or reconstructing pixels (He et al., 2022). This paradigm originated in the natural language processing (NLP) domain (Devlin et al., 2018; Radford et al., 2019) and enforces the learning of contextual reasoning and generative skills. Most applications in the image domain, usually named Masked Image Modeling (MIM), define low-level image information, such as pixels (He et al., 2022; Xie et al., 2022), as prediction targets. Although they

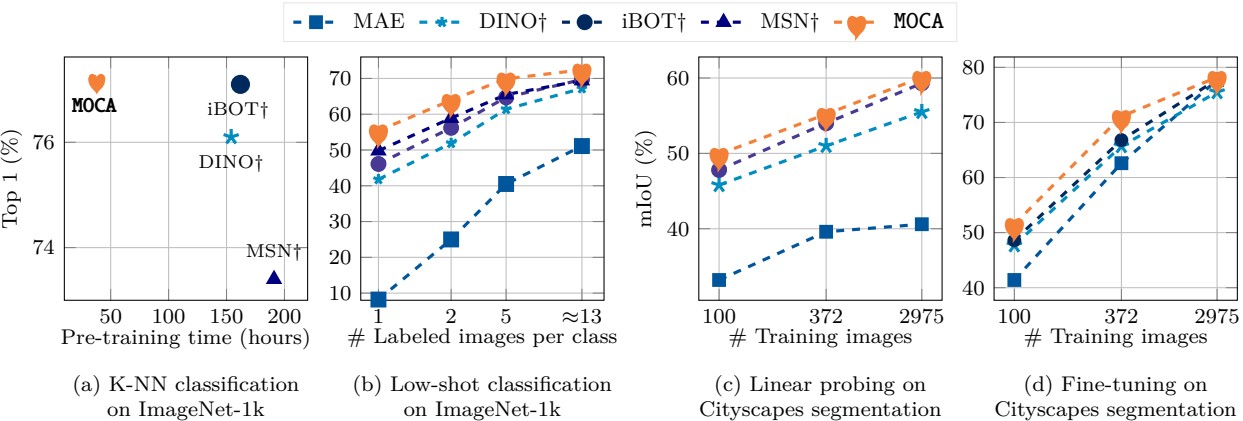

Figure 1: **Comparison of `MOCA` with state-of-the-art methods using ViT-B/16.** (a) K-NN ImageNet classification accuracy *vs*. pre-training time; (b) One-, two-, five- and ≈13-shot (the last corresponding to 1% training data) ImageNet classification accuracy; (c) Semantic segmentation on Cityscapes using linear probing and (d) fine-tuning with 100, 372, and 2975 training images. `MOCA` achieves superior results whilst requiring 3 times less training time. '†' denotes usage of multiple crops (Caron et al., 2021).

reach strong results when fine-tuning on a downstream task with *sufficient* training data (Bao et al., 2022; He et al., 2022), they do not offer *high-level "ready-to-use"* representations, typically evaluated in linear probing and k-NN classification settings. This is partly because predicting low-level pixel details is not conducive to learning representations that are "readily" available for downstream semantic tasks

Additionally, they do not promote the learning of invariances to image perturbations, which is an important characteristic of good image representations. This is actually what several methods –which are *discriminative*– have been designed to learn via contrastive- (Chen et al., 2020a; He et al., 2020; Misra & Maaten, 2020), teacher-student- (Grill et al., 2020; Gidaris et al., 2021; Assran et al., 2022; Caron et al., 2021) or clustering-style (Caron et al., 2018; Asano et al., 2020; Caron et al., 2020) objectives. However, such methods do not necessarily promote contextual reasoning and generative skills, crucial for effective visual representations, as the hide-and-predict methods more explicitly do.

Motivated by the above observations, we propose a dual self-supervised learning strategy that exhibits the learning principles of both discriminative and hide-and-predict paradigms. To accomplish this non-trivial goal, we propose a novel *masking-based method*, named `MOCA`, in which the two learning paradigms are defined in the *same space of high-level features*. In particular, our masking-based method enforces the good "reconstruction" of patch-wise codebook assignments that encode high-level and perturbation invariant features. Those assignment vectors are produced in an online and standalone fashion (i.e., without requiring pre-trained models or multiple training stages) using a *teacher-student* scheme and an *online generated codebook*. Indeed, the teacher produces spatially-dense codebook assignments from the unmasked image views while the student is trained to predict these assignments with masked views. Our masked-based training framework integrates in a unified way both a *dense patch-wise (local) loss* that encourages detailed feature generation and a *global image-wise loss* that enforces consistent image representation across different views–masked or not–of the same original image. As a result, our method converges quickly, reaching state-of-the-art performances with a training at least three times faster than the one of prior methods (see Fig. 1(a)).

To summarize, our contributions are: **(1)** We propose a self-supervised representation learning approach called `MOCA` (for Masked Online Codebook Assignments prediction) that unifies both perturbation invariance and dense contextual reasoning. As we will show, combining these two learning principles in an effective way is not a trivial task. **(2)** We demonstrate that `MOCA` achieves state-of-the-art results in low-shot settings (e.g., see Fig. 1(b-d)) and strong results when using the learned representation as an initialization for full fine-tuning, as well as for linear or k-NN classification settings, for which pixel reconstruction approaches under-perform. Finally **(3)**, our approach consists of a single end-to-end training stage that does not require prior pre-trained models. At the same time, `MOCA` is able to learn good representations with a much smaller training computation budget than competing approaches (see Fig. 1(a)).

## 2 Related Work

**Vision transformers.** State-of-the-art in NLP tasks (Devlin et al., 2018), Transformers (Vaswani et al., 2017) have been recently adapted to vision (Dosovitskiy et al., 2021; Touvron et al., 2021a; Zhao et al., 2020). The cadence to which tricks (Liu et al., 2021; Touvron et al., 2021b; Yuan et al., 2021) and architecture improvements (Touvron et al., 2021a; Zhai et al., 2022b) have been published has allowed for Transformer models to now also be high-ranked in most vision tasks (Carion et al., 2020; Liu et al., 2021; Touvron et al., 2022). However powerful the architecture is, fully-supervised training of Transformer models has proven to be non-trivial (Liu et al., 2020; Huang et al., 2020; Xiong et al., 2020), requiring extremely large amounts of annotated data (Dosovitskiy et al., 2021) and tuning, more than for the equivalent in size CNNs, thus making the learning of downstream tasks difficult. At the same time, unlike CNNs, ViTs do not easily saturate with more training data, making them particularly interesting for training on large collections of unlabelled images in a self-supervised manner (Bommasani et al., 2021).

**Self-supervised learning.** Self-supervised learning has emerged as an effective strategy for learning visual representations from unlabelled data without any human manual annotations. In its most common form, a model is trained on an annotation-free pretext task, e.g., (Doersch et al., 2015; Gidaris et al., 2018; Larsson et al., 2016; Misra et al., 2016; Noroozi & Favaro, 2016; Pathak et al., 2016), and then fine-tuned on a downstream task of interest with fewer annotated samples. The rise of *contrastive objectives* (Bachman et al., 2019; Dosovitskiy et al., 2014; Oord et al., 2018; Wu et al., 2018) has finally shown that self-supervised models can surpass the performance of supervised models on downstream tasks. These methods contrast representations between similar and dissimilar views, relying on well-thought data augmentations to create positive samples (Chen et al., 2020a;b; He et al., 2020; Misra & Maaten, 2020; Tian et al., 2020). Other forms of contrastive learning rely on clustering (Asano et al., 2020; Caron et al., 2018; 2019; 2020; Gidaris et al., 2020) or on various forms of self-distillation (Bardes et al., 2022; Caron et al., 2021; Chen & He, 2021; Gidaris et al., 2021; Grill et al., 2020; Zbontar et al., 2021) by learning to predict the similar representations for different views of the same image. Recently, a number of these methods have been successfully adapted to ViT architectures (Assran et al., 2022; Caron et al., 2021; Chen et al., 2021). Such *discriminative* methods do not perform spatial reasoning. In this work, we propose to facilitate learning by also enforcing local reasoning with an objective inspired by masked image modeling methods which we describe below.

**Masked image modeling.** The success of transformers in NLP has also been enabled by effective self-supervised pre-training tasks such as mask autoencoding in BERT (Devlin et al., 2018) or language-modeling in GPT (Radford & Narasimhan, 2018; Radford et al., 2019). Translated in the vision paradigm as Masked Image Modeling (MIM), such tasks have rapidly gained popularity through their simplicity and their direct compatibility with ViTs processing images as sequences of patch tokens (Bao et al., 2022; Chen et al., 2022; He et al., 2022; Xie et al., 2022; Li et al., 2021; Zhou et al., 2022; Fang et al., 2023). MIM approaches come with different reconstruction targets for the masked input tokens: RGB pixels (He et al., 2022; Xie et al., 2022), hand-crafted HOG descriptors (Wei et al., 2022a) or token features computed by a teacher network (Assran et al., 2022; Baevski et al., 2022; Bao et al., 2022; El-Nouby et al., 2021; Li et al., 2021; Dong et al., 2023; Zhou et al., 2022). For the latter, most of the methods rely on a static vocabulary of tokens generated by a pre-trained generative model, e.g., VQ-VAE (Ramesh et al., 2021), where the MIM task consists in reconstructing the tokens corresponding to the masked patches (Bao et al., 2022; Dong et al., 2023; Li et al., 2022; Peng et al., 2022). Recent methods resort to higher capacity teachers like CLIP (Radford et al., 2021) leading to significant performance boosts (Hou et al., 2022; Peng et al., 2022; Wei et al., 2022b;c; Xue et al., 2023). Although convenient, this strategy involves multiple training stages, including the computationally expensive training of the visual tokenizer. Alternatively, the target tokens can be computed online in a *self-distillation*-like manner (Assran et al., 2022; Baevski et al., 2022; Kakogeorgiou et al., 2022; Zhou et al., 2022). However, since these ViT-based models essentially start from scratch, they may require many epochs to bootstrap the learning of token targets and of rich representations. `MOCA` efficiently follows this approach, being at least three times faster than current methods. Our scheme trains the student to densely predict teacher token assignments of masked patches over an online generated codebook. Other works quantize tokens into static pre-computed codebooks (Peng et al., 2022) or predict assignments of global image features over an online codebook (Assran et al., 2022; Caron et al., 2020), though potentially losing spatial information.

# 3 Our approach

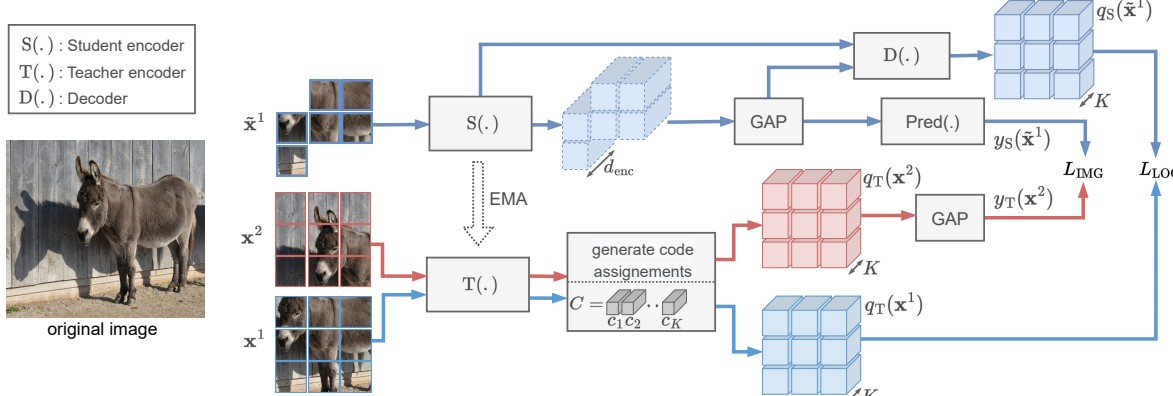

Figure 2: **Overview of MOCA .** The teacher (bottom) takes as an input two unmasked random views $\mathbf{x}^{\{1,2\}}$ of the same image and generates dense token-wise code assignments $q_T(\mathbf{x})$ for them (i.e., soft-assigns codebook items to the patch tokens). The student (top) receives as an input a randomly masked image version $\tilde{\mathbf{x}}^1$ of view $\mathbf{x}^1$ and is trained to minimize two types of self-supervised losses: (1) A *masked same-view token assignment prediction* loss $L_{\text{LOC}}$, which requires predicting the teacher-produced assignment vectors of view $\mathbf{x}^1$ from the corresponding masked image $\tilde{\mathbf{x}}^1$. This is a spatially dense loss that enables learning representations with dense contextual reasoning. (2) A *masked cross-view average assignment prediction* loss $L_{\text{IMG}}$, which is to predict with the global image embedding of the first view $\tilde{\mathbf{x}}^1$ the average assignment vector of the opposite view $\mathbf{x}^2$ ('GAP' stands for Global Average Pooling). This is an image-wise loss that promotes learning image representations that are invariant with respect to different augmentations of the input. The same objectives are applied in a symmetric way when the student gets as input the masked image version $\tilde{\mathbf{x}}^2$ of view $\mathbf{x}^2$ (not shown). We implement the $L_{\text{LOC}}$ objective with a condenser-based decoder that gets as input patch-token embeddings from an intermediate layer of the student encoder and the global image embeddings from its last layer (see Sec. 3.3). The teacher is an exponential moving average ('EMA') of student.

Our goal is to build a hide-and-predict self-supervised approach that (a) is defined over high-level concepts and (b) is standalone, i.e., does not require any pre-trained model or off-line training stage. To this end, we follow a teacher-student setup where the teacher transformer $T(\cdot)$ is a momentum-updated version (He et al., 2020; Richemond et al., 2020) of the student transformer $S(\cdot)$ and generates targets for its student (Figure 2).

For defining these targets, we use what we will refer to hereafter as *token assignment vectors* (over an online codebook), which constitute the high-level visual concepts over which our hide-and-predict framework is defined. We detail the definition of these assignment vectors in Sec. 3.1. We then describe in Sec. 3.2 how we use these teacher-produced assignment vectors for defining the hide-and-predict objectives for the student training. In Sec. 3.3 we discuss important design choices and in Sec. 3.4 techniques for faster training.

## 3.1 Teacher-produced token assignment vectors

Given as input an image token sequence $\mathbf{x} = \{\mathbf{x}_i\}_{i=1}^N$, the teacher transformer first extracts $d_{\text{enc}}$-dimensional token embeddings $T(\mathbf{x}) = \{T(\mathbf{x})_i\}_{i=1}^N$, and then soft-quantizes them over a codebook $C = [\mathbf{c}_1, \ldots, \mathbf{c}_K]$ of $K$ embeddings of dimension $d_{\text{enc}}$. This produces the soft-assignment vectors $q_T(\mathbf{x}) = \{q_T(\mathbf{x})_i\}_{i=1}^N$, where $q_T(\mathbf{x})_i$ is a $K$-dimensional vector computed as the cosine similarity of the token embedding $T(\mathbf{x})_i$ and each codebook embedding $\mathbf{c}_k$, followed by a softmax with temperature $\tau_T$, which controls the softness of the assignment. In essence, $q_T(\mathbf{x})_i$ encodes the assignment of $T(\mathbf{x})_i$ to its closest (in terms of cosine distance) codebook embeddings. We use these teacher-produced assignment vectors for defining the hide-and-predict objectives for the student training, which is described in Sec. 3.2. For the softmax temperature we use $\tau_T = \frac{1}{10 \cdot \bar{\mu}_{\text{MSD}}}$, where $\bar{\mu}_{\text{MSD}}$ is the exponential moving average (with momentum 0.99) of per-token difference between the maximum and average cosine similarity over the codebook embeddings. For the codebook, we use $K = 4096$.

**Simple online codebook construction.** Following Gidaris et al. (2021), we build the codebook $C$ as a queue of size $K$ containing teacher-token embeddings randomly sampled from previous mini-batches. During each training step, after computing assignment vectors, we randomly sample $K_{\text{new}} \ll K$ token embeddings from the current mini-batch, each from a different image. Specifically, we uniformly select images and then randomly choose (again with uniform distribution) one teacher patch-token embedding from each. These selected embeddings replace the oldest ones in the queue, keeping only those from the past $K/K_{\text{new}}$ mini-batches in memory. Codebook embeddings are initially randomly initialized from a normal distribution at training step 0. Since the learning rate is very low during the initial training steps (due to the warmup period), the random initialization of the embeddings has minimal impact on the learning process in the early stage.

### 3.2  Hide-and-predict token assignment vectors

Let $\mathbf{x}^v$ with $v \in \{1, 2\}$ be the token sequences of two random views sampled from the same training image. On the teacher side, they are processed by the transformer $\text{T}(\cdot)$ to produce a sequence of image token assignment vectors $q_\text{T}(\mathbf{x}^v)$. On the student side, the transformer $\text{S}(\cdot)$ receives as input randomly masked versions of the $\mathbf{x}^v$ images, which are denoted by $\tilde{\mathbf{x}}^v$ with $v \in \{1, 2\}$, and is trained to minimise two types of masked-based self-supervised losses:

1. A *masked same-view token assignment prediction* loss $L_{\text{LOC}}$, which aims at predicting the sequence of teacher assignment vectors $q_\text{T}(\mathbf{x}^v)$ from the corresponding masked image $\tilde{\mathbf{x}}^v$ for $v \in \{1, 2\}$. This is a spatially dense loss whose role is to enable learning representations with dense contextual reasoning skills.

2. A *masked cross-view average assignment prediction* loss $L_{\text{IMG}}$, which amounts to predicting the teacher's average assignment vector of the first view, $y_\text{T}(\mathbf{x}^1)$, from the opposite masked view $\tilde{\mathbf{x}}^2$ and vice-versa. This is an image-wise loss whose role is to promote learning representations that are invariant with respect to different input augmentations.

The overall training loss for the student model is:

$$L = \lambda L_{\text{IMG}} + (1 - \lambda) L_{\text{LOC}}, \tag{1}$$

where $\lambda \in [0, 1]$. Unless stated otherwise, we use $\lambda = 0.5$. Next, we describe these two objectives in detail.

**Masked same-view token assignment prediction.** With this objective, given as input the masked view $\tilde{\mathbf{x}}^v$, the student is trained to predict the assignment vectors of the tokens as generated by the teacher from $\mathbf{x}^v$. In the following, for each $v \in \{1, 2\}$, we denote by $u_v \subset \{1, ..., N\}$ the set of indices of all tokens that remain unmasked (i.e., all tokens from view $\mathbf{x}^v$ that are seen by the student) and by $m_v \subset \{1, ..., N\}$ the set of indices of all the masked tokens (it holds $m_v \cup u_v = \{1, ..., N\}$ and $m_v \cap u_v = \emptyset$)[1].

For computational efficiency reasons, the student encoder $S(\cdot)$ processes only the unmasked tokens. Therefore, for the prediction of the masked token assignment vectors, we employ an additional decoder transformer $\text{D}(\cdot)$. For this prediction to take place, *in the default setup*, we employ the following steps:

**Encoding the masked view:** We first give the unmasked tokens $\tilde{\mathbf{x}}^v = \{\mathbf{x}^v_i\}_{i \in u_v}$ as input to the student encoder transformer $\text{S}(\cdot)$ to produce the $d_{\text{enc}}$-dimensional student token embeddings $\{\text{S}(\tilde{\mathbf{x}}^v)_i\}_{i \in u_v}$.

**Decoder input embedding:** Before we give the encoder embeddings as input to the decoder, we apply to them a linear projection layer to convert their dimensionality from $d_{\text{enc}}$ to $d_{\text{dec}}$, where $d_{\text{dec}}$ is the dimensionality of the decoder embeddings. Then, we append the encoder embeddings with a learnable masked embedding for each masked token $i \in m_v$ and add (non-learnable sin-cos-based) positional embeddings to them. We denote the set of embeddings produced from this step as $\mathbf{z}^v$.

**Local token assignment prediction:** Last, the above embeddings are given as input to the decoder transformer $\text{D}(\cdot)$. The decoder computes the tokens embeddings $\text{D}(\mathbf{z}^v) = \{\text{D}(\mathbf{z}^v)_i\}_{i=1}^N$, which will be used

---

[1]We always assume that the [CLS] token is assigned the index 0.

for predicting the token assignment vectors as follows:

$$q_S(\tilde{\mathbf{x}}^v)_i = \text{softmax}(\frac{1}{\tau_d} \text{D}(\mathbf{z}^v)_i^\top W^d), \tag{2}$$

for $i \in \{1, ..., N\}$. The weights $W^d = [\mathbf{w}_1^d, \cdots, \mathbf{w}_K^d]$ are $L_2$-normalized $d_{\text{dec}}$-dimensional prototypes (one per codebook embedding in $C$) used for predicting the assignment vectors. As we explain in more detail in Sec. 3.2.1, these prototypes are not learnt but are dynamically generated by a weight generation module. Also, $\tau_d$ is a softmax temperature that controls the softness of the predicted probability.

The masked same-view token assignment prediction loss is

$$L_{\text{LOC}} = L_{\text{LOC}}^1 + L_{\text{LOC}}^2, \text{ where} \tag{3}$$

$$L_{\text{LOC}}^v = \sum_i \text{CE}\big(q_S(\tilde{\mathbf{x}}^v)_i, q_T(\mathbf{x}^v)_i\big), \forall v \in \{1, 2\}, \tag{4}$$

and $\text{CE}(\mathbf{a}, \mathbf{b}) = -\sum_{k=1}^K \mathbf{b}[k] \log(\mathbf{a}[k])$ is the cross-entropy loss between two distributions $\mathbf{a}$ and $\mathbf{b}$.

**Masked cross-view average assignment prediction.** In this objective, we first reduce the teacher assignments $q_T(\mathbf{x}) = \{q_T(\mathbf{x})_i\}_{i=1}^N$ to a single $K$-dimensional vector $y_T(\mathbf{x})$ via global average pooling, i.e., we reduce $q_T(\mathbf{x})$ to a 'bag-of-word' (BoW) representation. Then, similar to BoW-prediction methods (Gidaris & Komodakis, 2019; Gidaris et al., 2021), for each view $v$ the student must predict with its global image embedding[2] $\bar{\text{S}}(\tilde{\mathbf{x}}^v)$ the reduced assignment vector of the opposite view as follows:

$$y_S(\tilde{\mathbf{x}}^v) = \text{softmax}(\frac{1}{\tau_b} \bar{\text{S}}(\tilde{\mathbf{x}}^v)^\top W^b). \tag{5}$$

As in the same-view token assignment prediction, $\tau_b$ is the softmax temperature, and $W^b = [\mathbf{w}_1^b, \cdots, \mathbf{w}_K^b]$ are $L_2$-normalized $d_{\text{enc}}$-dimensional prototypes that are dynamically generated by a weight generation module.

The cross-view image-wise loss that is minimized for the two masked image views $\tilde{\mathbf{x}}^v$ with $v \in \{1, 2\}$, is

$$L_{\text{IMG}} = \text{CE}\big(y_S(\tilde{\mathbf{x}}^1), y_T(\mathbf{x}^2)\big) + \text{CE}\big(y_S(\tilde{\mathbf{x}}^2), y_T(\mathbf{x}^1)\big). \tag{6}$$

### 3.2.1 Assignment predictions using dynamic prototype generation modules

Our method updates the codebook $C$ at the teacher side at each training iteration with new randomly-sampled teacher-token embeddings. Due to this, instead of learning the prototypes $W^d$ and $W^b$ that are used on the student side for the two assignment prediction tasks defined by Eqs. (2) and (5), we dynamically generate them using two weight generation modules. In particular, we employ the generation networks $\text{G}^b(\cdot)$ and $\text{G}^d(\cdot)$ that at each training step take as input the current codebook $C = [\mathbf{c}_1, \ldots, \mathbf{c}_K]$ and produce for them the prototypes weights $W^b = \text{G}^b(C) = [\text{G}^b(\mathbf{c}_1), \ldots, \text{G}^b(\mathbf{c}_K)]$ and $W^d = \text{G}^d(C) = [\text{G}^d(\mathbf{c}_1), \ldots, \text{G}^d(\mathbf{c}_K)]$ respectively. The networks $\text{G}^b(\cdot)$ and $\text{G}^d(\cdot)$ are implemented with 2-layer perceptrons (i.e., each as a sequence of Linear, BN, ReLU, and Linear layers), whose input and output vectors are $L_2$-normalized and have a skip connection between them (more implementation details in Appendix C.1).

We use $\tau_b = \tau_d = \frac{1}{3}$ as the softmax temperatures in Eqs. (2) and (5) respectively.

### 3.3 Learning high-level target assignment vectors

For our hide-and-predict approach to be effective, the teacher must learn assignment vectors encoding high-level visual concepts. In the context of our teacher-student scheme, where the teacher is a slower-moving version of the student, this depends on the student first learning patch-token embeddings (from where the assignment vectors are extracted at the teacher side) that capture this type of visual features. Below we detail the critical role some design choices have in that.

---

[2]As global image embedding we use the average token embedding $\bar{\text{S}}(\tilde{\mathbf{x}}^v) = \frac{1}{|u_v|} \sum_{i \in u_v} \text{S}(\tilde{\mathbf{x}}^v)_i$. We found this to work better than using the [CLS] token embedding $\text{S}(\tilde{\mathbf{x}}^v)_0$.

Table 1: **Training efficiency.** (a) Training savings and (b) k-NN results on ImageNet-1k with ViT-Base/16 models pre-trained for 100 epochs. 'Partial Dec.': partial decoding; '#Masks': number of different masks per view; 'k-NN': top-1 accuracy (%); 'Time' and 'Memory': per epoch training time and GPU memory footprint measured with a single 8-A100 node and batch size 2048.

(a) **Training savings** by switching full to partial decoding.

| Decoder depth | Time (min) | Memory (GB) |
|---|---|---|
| 2 | $9.00 \rightarrow 8.15$ | $30.4 \rightarrow 23.0$ |
| 4 | $10.25 \rightarrow 8.55$ | $37.7 \rightarrow 26.2$ |
| 8 | $12.41 \rightarrow 9.30$ | $52.4 \rightarrow 32.6$ |

(b) **Partial decoding & 2nd masking round.** The decoder depth is 2.

| Partial Dec. | #Masks | k-NN (%) | Time (min) | Memory (GB) |
|---|---|---|---|---|
| ✗ | 1 | 71.6 | 9.00 | 30.4 |
| ✓ | 1 | 71.8 | 8.15 | 23.0 |
| ✓ | 2 | 74.4 | 10.35 | 39.5 |

**Condenser-based decoding.** As in Gao & Callan (2021); Peng et al. (2022), we change the default setup described in Sec. 3.2 and create a bottleneck in the decoding task that promotes the student's global image representation to capture more spatial structure from the input view $\tilde{\mathbf{x}}$, crucial for predicting patch token assignments. The bottleneck is introduced by changing the way the decoder input embeddings $\mathbf{z}^v$ are formed.

In particular, instead of using as input to the decoder the token patch embeddings $\{S(\tilde{\mathbf{x}}^v)_i\}_{i \in u_v}$ (coming from its last layer), we give as input **(a)** token embeddings $\{S^\ell(\tilde{\mathbf{x}}^v)_i\}_{i \in u_v}$ from an intermediate transformer layer $\ell$ of S together with **(b)** the last layer's global image embedding $\bar{S}(\tilde{\mathbf{x}}^v)$. Using lower-layer embeddings for the patch tokens introduces a bottleneck. To compensate, the decoder must rely more on the global image embedding $\bar{S}(\tilde{\mathbf{x}}^v)$ that it receives as input. This forces the global image embedding to capture additional spatial structure information, potentially beneficial for downstream tasks such as k-NN and linear probing.

**Hide-and-predict using the [AVG] token.** The average patch embedding $\bar{S}(\tilde{\mathbf{x}}^v) = \frac{1}{|u_v|} \sum_{i \in u_v} S(\tilde{\mathbf{x}}^v)_i$ (called [AVG] token for brevity) is used for performing **(a)** the cross-view image-wise prediction task (see Eq. (5)), which is crucial for promoting perturbation invariances, as well as **(b)** the condenser-based decoding that was explained above. Therefore, using the [AVG] token (instead of the [CLS] token) results in the student's patch embeddings $\{S(\tilde{\mathbf{x}}^v)_i\}_{i \in u_v}$ (from which the [AVG] token is computed) to receive a more direct supervisory signal from the two prediction losses. Consequently, these embeddings (i) are steered to encode the perturbation-invariant and context/structure-aware information necessary for performing the two prediction objectives, and (ii) are typically of a more semantic nature.

The impact of these design choices is studied in Sec. 4.1.

### 3.4 Masking strategies for efficient training

**Partial decoding for efficient training.** We randomly select the token indices $m_v$ to be masked from each view, with a relatively high percentage of masked tokens (i.e., 65%). This results in significant computational and GPU memory savings from the student encoder side during training. However, some of these savings are lost if the decoder processes all masked tokens. To mitigate this, we propose inputting only a small subset $m'_v \subset m_v$ of masked tokens, along with the visible tokens, to the decoder. For example, instead of inputting all 65% of image tokens that we mask from the student, we input only a random subset, e.g., 20% of the total image tokens. As shown in Tab. 1a, this results in noticeable savings in computation and GPU memory, especially for many-layer decoders, without any loss in the quality of learned representations (as we demonstrate in Sec. 4.1; see Tab. 1b). Note that although we only tested partial decoding in the context of our approach, we believe it could benefit other hide-and-predict methods as well, e.g., MAE (He et al., 2022).

**Faster training convergence with 2nd masking round.** With our partial decoding, we can allocate our computation budget to enhance the model's performance with other techniques. Specifically, we apply a second round of random token masking to both views and then perform the assignment prediction objectives with the newly generated masked views in the same manner as before. This technique artificially increases the batch size and saves time since we can reuse the teacher assignment vectors from the first masking round. It

Table 2: `MOCA` **ablation.** k-NN classification (%) on ImageNet-1k with a ViT-B/16 trained for 100 pre-training epochs and one masking round. Default setting (in gray): $\lambda = 0.5$, mask ratio of 65%, [AVG] token used as global image embedding, condenser-based decoder with depth 2 fed with patch-embeddings from the $\ell = 8$ intermediate student layer. 'Time' and 'Memory': per epoch training time (min) and GPU memory footprint (GB) measured with a single 8-A100 node and batch size 2048.

(a) **Condenser-based decoding** is more effective.

| Objectives | $L_{\text{IMG}}$ ($\lambda$=1.0) | $L_{\text{IMG}}$ & $L_{\text{LOC}}$ ($\lambda$=0.5) | |
|---|---|---|---|
| Condenser | N/A | ✗ | ✓ |
| k-NN | 66.8 | 67.8 | **71.8** |

(b) **Decoder depth**. "Shallower" is more accurate and faster.

| Depth | 1 | 2 | 4 | 8 |
|---|---|---|---|---|
| k-NN | 71.7 | **71.8** | 70.9 | 70.5 |
| Time | 8.08 | 8.15 | 8.55 | 9.3 |
| Memory | 21.5 | 23.0 | 26.2 | 32.6 |

(c) **Masking ratio**. The values 55% and 65% give the best results.

| Masking | 55% | 65% | 75% | 80% |
|---|---|---|---|---|
| k-NN | **72.6** | 71.8 | 69.8 | 68.4 |
| Time | 9.00 | 8.15 | 7.37 | 7.35 |
| Memory | 28.6 | 23.0 | 18.2 | 15.7 |

(d) [**CLS**] **vs.** [**AVG**] as global image embedding.

| im. embed. | [CLS] | [AVG] |
|---|---|---|
| k-NN | 60.2 | 71.8 |

(e) **Weight** $\lambda$ in $\lambda L_{\text{IMG}} + (1-\lambda)L_{\text{LOC}}$. Both objectives are important.

| $\lambda$ | 1.0 | 0.75 | 0.5 | 0.25 | 0.00 |
|---|---|---|---|---|---|
| k-NN | 66.8 | 70.2 | **71.8** | 71.5 | 13.1 |

(f) **Intermediate layer** $\ell$ in condenser-based decoding.

| Layer $\ell$ | 6 | 8 | 9 | 10 |
|---|---|---|---|---|
| k-NN | 71.4 | **71.8** | 71.5 | 70.9 |

can be seen as a form of batch augmentation (Hoffer et al., 2020; Touvron et al., 2022) that reduces gradient variance by having in the same mini-batch multiple randomly augmented copies of the same sample.

## 4 Experiments

**Setup.** We evaluate our `MOCA` method by training ViT-B/16 models on the ImageNet-1k (Russakovsky et al., 2015) dataset. We use the AdamW optimizer (Loshchilov & Hutter, 2019) with $\beta_1 = 0.9$, $\beta_2 = 0.999$ and weight decay 0.05. The batch size is 2048 split over 8 A100 GPUs. For the learning rate $lr$, we use a linear warm-up from 0 to its peak value for 30 epochs and then decrease it over the remaining epochs with a cosine annealing schedule. The peak $lr$ is $1.5 \times 10^{-4}$ and the number of training epochs is 100 or 200. More implementation details are provided in the appendix.

### 4.1 Method analysis

We first study several aspects of our approach with ablative experiments, in which we train ViT-B/16 models for 100 epochs. Except stated otherwise, the models include the condenser-based decoder (Sec. 3.3), partial decoding, and a single token masking round. We evaluate the learned representations on the k-NN ImageNet classification task.

**Integrating the two masked prediction objectives.** Here we show that a naive integration of the two masked-prediction objectives does not work effectively.

*Ablating the condenser-based decoding in Tab. 2a.* Compared to a model trained only with the image-wise objective $L_{\text{IMG}}$, a naive implementation of our dense hide-and-predict objective $L_{\text{LOC}}$ without the condenser design principle provides only a small k-NN performance improvement (i.e., +1 point). Instead, when using the condenser, the k-NN performance increase is much larger (i.e., +5 points). We further compare the performance with and without condenser when pre-training for 200 epochs (instead of 100) in Tab. 3: condenser leads to significantly better results in all evaluation protocols, apart from ImageNet fine-tuning. On the other hand, in this longer pre-training setting the naive implementation of $L_{\text{LOC}}$ achieves worse results than only using $L_{\text{IMG}}$ in k-NN and linear probing. These results validate that the condenser design choice indeed leads to a more effective hide-and-predict training (see discussion in Sec. 3.3). In Tab. 2f we see that using $\ell = 8$ as the intermediate student layer for providing the patch-token embeddings that are given as input to the condenser-based decoder, gives the best results.

Table 3: **Ablating hide-and-predict variants.** ImageNet-1k classification results with ViT-B/16 models pre-trained for 200 epochs with one masking round. The improvements are over the $L_{\mathrm{IMG}}$-only objective.

| Target Objectives | $L_{\mathrm{IMG}}$ | +hide-and-predict variants | | | |
|---|---|---|---|---|---|
| | | Assign. | Assign. (ours) | Pixels | Embeddings |
| **Condenser** | N/A | ✗ | ✓ | ✗ | ✓ |
| **k-NN** | 72.0 | 70.0 (−2.0) | **75.4 (+3.4)** | 72.3 (+0.3) | 73.6 (+1.6) |
| **Linear** | 76.1 | 75.7 (−0.4) | **77.7 (+2.0)** | 76.4 (+0.3) | 76.9 (+1.2) |
| **Fine-tune** | 82.7 | **83.5 (+0.8)** | 83.4 (+0.7) | 83.2 (+0.5) | 83.0 (+0.3) |

*[CLS] vs. [AVG] global embedding tokens (Tab. 2d).* Using the [CLS] token as the global embedding, instead of the average embedding token ([AVG]), significantly decreases the k-NN performance (i.e., −11.6 points). We postulate that this is because, as discussed in Sec. 3.3, the [AVG] token leads to learning assignment vectors that encode more high-level concepts and thus makes our hide-and-predict method more effective.

**Balancing the two prediction objectives.** The $\lambda$ hyperparameter controls the importance of the two assignment prediction objectives (see Eq. (1)). We see in Tab. 2e that $\lambda = 0.5$ gives the best results while switching off any of the two objectives ($\lambda = 1$ or $\lambda = 0$) seriously deteriorates the k-NN performance. This suggests that the objectives work in a complementary and synergistic way. For instance, for $\lambda = 0$ (no $L_{\mathrm{IMG}}$ loss) the results are very poor. The reason is that without the image-wise loss, it is hard to bootstrap the learning of teacher-produced assignment vectors that encode high-level concepts (see discussion in Sec. 3.3). In the appendix, we report more experiments, results, and discussions about the behavior when training without the image-wise loss $L_{\mathrm{IMG}}$.

**Decoding and masking strategies.**

*Studying the impact of decoder depth in Tab. 2b.* We observe that our method behaves better with "shallower" decoders, in contrast to what is the case in MAE (which uses a decoder with eight layers). This is expected since our hide-and-predict objectives require the decoder to predict high-level concepts. In that case, it is better for the applied loss to be "closer" to the student (i.e., having a shallower decoder) so as to promote the output student embeddings to capture such high-level information in a more "ready-to-use" way. On the other hand, MAE's decoder, which must reconstruct pixels, is better to be deeper and thus spare the MAE's encoder from capturing such low-level details. Furthermore, using a "shallower" decoder leads to significant computation and GPU memory savings in our case.

*Partial decoder and 2nd masking round in Tab. 1b.* Switching off partial decoding (i.e., decoding all the masked tokens) leads to an increase in the per-epoch training time and GPU memory usage with zero benefits on the quality of the learned features as measured with k-NN. Instead, having a second round of token masking, although increasing the training time and GPU memory usage, leads to a significant k-NN performance boost. Therefore, we employ it for the final model that we train for the results in Sec. 4.4.

*Studying the masking ratio impact in Tab. 2c.* The 55% and 65% ratios give the best results.

## 4.2 Comparison of hide-and-predict objectives

In our work, we propose a hide-and-predict objective $L_{\mathrm{LOC}}$ that is defined in the space of assignment vectors over an online-updated codebook. In Tab. 3, we compare this objective ('Assign.') against **(a)** a hide-and-predict objective defined in the pixel space with per-patch normalization as in MAE (He et al., 2022) ('Pixels'); and **(b)** a hide-predict-objective in which the decoder must directly regress the $d_{\mathrm{enc}}$-dimensional output teacher-token embeddings (before any code assignment) using a cosine-distance loss ('Embeddings'). We keep the image-wise $L_{\mathrm{IMG}}$ objective intact for all three models. The Embeddings model is implemented with a condenser-based decoder of depth two, as with our Assignments model, while the Pixels model uses a

Table 4: **Ablations on Cityscapes semantic segmentation with linear probing.** We report mIoU (%) results with ViT-B/16 models pre-trained for 200 epochs. For training we use 100 or 372 images from the Cityscapes training set using random splits from (French et al., 2020).

| | | | Linear probing | |
|---|---|---|---|---|
| Objectives | Condenser | #Masks | 100 | 372 |
| $L_{IMG}$ | N/A | 1 | 44.6 | 50.9 |
| $L_{IMG}$ & $L_{LOC}$ | ✗ | 1 | 44.2 | 51.5 |
| $L_{IMG}$ & $L_{LOC}$ | ✓ | 1 | 46.8 | 52.4 |
| $L_{IMG}$ & $L_{LOC}$ | ✓ | 2 | **49.9** | **55.2** |

standard decoder[3] and depth four (which gives better results). All other model hyper-parameters remain the same for fair comparison. We see that the Assignments model produces superior results on all three ImageNet classification evaluation protocols. This demonstrates the advantage of defining a hide-and-predict objective with online codebook assignment vectors as targets.

### 4.3 Ablations on low-shot semantic segmentation

Here we conduct ablation studies to evaluate (a) our hide-and-predict objective $L_{LOC}$, (b) the condenser decoder design, and (c) the second masking round, on the downstream task of low-shot Cityscapes semantic segmentation with linear probing; detailed information on this evaluation protocol can be found in Sec. 4.4 and Appendix C.3. The results confirming these design choices are presented in Tab. 4.

### 4.4 Comparative results

We compare our method against other hide-and-predict methods as well as self-supervised methods based on teacher-student or contrastive objectives. To this end, we train a ViT-B/16 model for 200 epochs with a condenser-based decoder, two masking rounds (the 1st with 55% mask ratio and the 2nd with 75% mask ratio), and partial decoding only 20% of patch tokens per masking round.

**Full and low-shot image classification.** We first evaluate the learned representations on ImageNet classification with the k-NN, linear probing, and fine-tuning evaluation protocols and using the full training set. Moreover, we evaluate low-shot ImageNet classification with logistic regression (Assran et al., 2022).

*Comparison with hide-and-predict approaches.* In Tab. 5a, we compare our method MOCA against other self-supervised methods with MIM objectives. We observe that on linear probing MOCA achieves superior performance to all the other works, with more than 2 points. This demonstrates that MOCA is better at learning "ready-to-use" features. At the same time, its fine-tuning performance is on par with the state-of-the-art, falling only 0.4 points behind the top-performing method. This is despite MOCA using only 200 epochs and a smaller computational budget in general. For instance, as we see in Tab. 5c, the total training time of MOCA is more than 3.5 times smaller compared to MAE, which is (one of) the most training-efficient methods.

*Comparison with teacher-student and contrastive methods.* In Tab. 5b, we compare our method against other self-supervised methods based on contrastive objectives or teacher-student schemes. We observe that our method is superior to competing approaches that do not use multiple crops and is on par with them when they employ such techniques. Again, we emphasize that our method is much more computationally efficient than these techniques (see Tab. 5c).

*Low-shot ImageNet-1k classification.* Here we adopt the low-shot evaluation protocol of MSN (Assran et al., 2022) and use as few as 1, 2, or 5 training images per class as well as using 1% of the ImageNet-1k's training data, which corresponds to ≈13 images per class. In particular, for this low-shot training setting, we extract global image embeddings with the pre-trained transformer (average patch embeddings computed by the

---

[3]We found that implementing the Pixels hide-and-predict version with a condenser-based decoder has worse performance than the model trained with only the image-wise $L_{IMG}$ objective.

Table 5: **Comparisons with prior methods.** ImageNet-1k classification and training time results with ViT-B/16 models. `MOCA`, pre-trained for 200 pre-training epochs and with two token masking rounds, is compared with **(a)** the *hide-and-predict* methods MaskFeat (Wei et al., 2022a), CAE (Chen et al., 2022), BEiT (Bao et al., 2022), SimMIM (Xie et al., 2022), MAE (He et al., 2022), CAN (Mishra et al., 2022), and **(b)** the *discriminative* methods MoCo-v3 (Chen et al., 2021), DINO (Caron et al., 2021), iBOT (Zhou et al., 2022) and MSN (Assran et al., 2022). In **(c)** we compare the per-epoch ('Epoch', in minutes) and total ('Total', in hours) training time. The per-epoch time measurements are based on a single 8 A100 GPU node. In **(d)** we evaluate using the low-shot image classification protocol. †: use of multiple crops (Caron et al., 2021). The k-NN and linear probing results of MSN were computed by us.

(a) **Hide-and-predict methods**

|  |  | top-1 accuracy (%) | |
| --- | --- | --- | --- |
| Method | #Epoch | Linear | F-tune |
| MaskFeat | 1600 | - | **84.0** |
| CAE | 1600 | 70.4 | 83.9 |
| BEiT | 800 | 37.6 | 83.2 |
| SimMIM | 800 | 56.7 | 83.8 |
| MAE | 1600 | 68.0 | 83.6 |
| CAN | 800 | 74.8 | 83.6 |
| `MOCA` (ours) | **200** | **78.7** | 83.6 |

(b) **Discriminative methods**

|  |  | top-1 accuracy (%) | | |
| --- | --- | --- | --- | --- |
| Method | #Epoch | k-NN | Linear | F-tune |
| MoCo-v3 | 300 | - | 76.7 | 83.0 |
| DINO | 400 | 68.9 | 72.8 | - |
| iBOT | 400 | 71.2 | 76.0 | - |
| DINO† | 400 | 76.1 | 78.2 | 82.8 |
| iBOT† | 400 | 77.1 | **79.6** | **84.0** |
| MSN† | 600 | 73.4 | 77.2 | 83.4 |
| `MOCA` (ours) | **200** | **77.2** | 78.7 | 83.6 |

(c) **Training time comparisons**

| Method | #Epoch | Epoch (mn) | Total (hr) |
| --- | --- | --- | --- |
| MAE | 1600 | 5.35 | 142 |
| DINO | 400 | 16.42 | 109 |
| iBOT | 400 | 17.40 | 116 |
| DINO† | 400 | 23.16 | 154 |
| iBOT† | 400 | 24.33 | 162 |
| MSN† | 600 | 19.07 | 191 |
| `MOCA` (ours) | 200 | 11.23 | 38 |

(d) **Low-shot ImageNet classification evaluation**

|  |  | #Images per class | | | |
| --- | --- | --- | --- | --- | --- |
| Method | #Epoch | 1 | 2 | 5 | ≈13 |
| MAE | 1600 | 8.2±0.3 | 25.0±0.3 | 40.5±0.2 | 51.1 |
| DINO† | 400 | 41.8±0.3 | 51.9±0.6 | 61.4±0.2 | 67.2 |
| iBOT† | 400 | 46.1±0.3 | 56.2±0.7 | 64.7±0.3 | 69.7 |
| MSN† | 600 | 49.8±0.2 | 58.9±0.4 | 65.5±0.3 | 69.4 |
| `MOCA` (ours) | **200** | **55.4±0.4** | **63.9±0.2** | **69.9±0.1** | **72.4** |

Table 6: **Semantic segmentation on Cityscapes.** We report mIoU (%) results using a Segmenter (Strudel et al., 2021) model with ViT-B/16 backbone. For training we use 100, 372 or 2975 (full training split) images from the Cityscapes training set using random splits from (French et al., 2020) for the 100- and 372-image settings. †: use of multi-crop augmentation during pre-training (Caron et al., 2021).

|  | Linear probing | | | Finetuning | | |
| --- | --- | --- | --- | --- | --- | --- |
| Method | 100 | 372 | 2975 | 100 | 372 | 2975 |
| MAE (He et al., 2022) | 33.2 | 39.6 | 40.6 | 41.4 | 62.6 | 77.7 |
| DINO† (Caron et al., 2021) | 45.8 | 51.0 | 55.5 | 47.7 | 65.7 | 75.7 |
| iBOT† (Zhou et al., 2022) | 47.8 | 54.0 | 59.3 | 48.6 | 66.8 | 77.8 |
| `MOCA` (ours) | **49.9** | **55.2** | **60.1** | **51.5** | **71.1** | **78.2** |

teacher) and train a linear classifier on top of it with the few available training data. As we see from the results in Tab. 5d (see also Fig. 1(b)), `MOCA` outperforms prior methods with ViT-B/16 – surpassing the second best method MSN (Assran et al., 2022) by a large margin.

**Full and low-shot semantic segmentation.** We further evaluate our method on the Cityscapes (Cordts et al., 2016) semantic segmentation dataset, by either linear probing or finetuning. We use the Segmenter (Strudel et al., 2021) model with ViT-B/16 backbone initialized either with our `MOCA` or other self-supervised methods. In the linear probing setup we append a linear layer to the frozen encoder and in finetuning setup we use a single mask transformer layer as a decoder following the original setup from Strudel et al. (2021). We train using the full Cityscapes training set of 2975 images as well as 100 or 374 training

images, representing 1/30 and 1/8 of the full training set. For these 100 and 374 low-shot settings, we use three different splits of 100 or 374 training images respectively following the protocol of French et al. (2020) and report the average mIoU performance over the three splits. Learning rates were optimized for every method separately, with other hyperparameters kept as the default ones used in the Segmenter (Strudel et al., 2021) codebase. We evaluate the trained segmentation models on the 500 Cityscapes validation images.

We report the linear probing results in Fig. 1(c) and fine-tuning results in Fig. 1(d) and provide detailed results in Tab. 6. They show that MOCA consistently achieves superior performance compared to prior methods in all the setups, with bigger improvements appearing in the 100 and 374 low-shot settings.

**COCO detection and instance segmentation results.**
We present results on COCO detection and instance segmentation in Tab. 7. MOCA demonstrates strong performance in these tasks. We use the COCO 2017 set consisting of 118K training images and 5k validation. For this experiment we follow the implementation of Liu et al. (2021). In detail, we adopt Cascade Mask R-CNN (He et al., 2017; Cai & Vasconcelos, 2019) as task layer and use a similar hyper-paramenter configuration for ViT-B with (Zhou et al., 2022): multi-scale training (resizing

Table 7: **COCO detection and instance segmentation with ViT-B/16.** †: use of multiple crops (Caron et al., 2021).

| Method | DINO† | iBOT† | MOCA |
|---|---|---|---|
| $\mathbf{AP}^{\mathrm{box}}$ | 50.1 | 51.2 | 50.5 |
| $\mathbf{AP}^{\mathrm{mask}}$ | 43.4 | 44.2 | 43.6 |

image with shorter size between 480 and 800, with the longer side no larger than 1333), AdamW (Loshchilov & Hutter, 2019) optimizer with initial learning rate $2e^{-4}$, the $1\times$ schedule (12 epochs with the learning rate decayed by $10\times$ at epochs 8 and 11).

## 5 Conclusion

In this work we have proposed a novel self-supervised learning framework that is able to exhibit the complementary advantages of both discriminative and hide-and-predict approaches, thus promoting visual representations that are perturbation invariant, encode semantically meaningful features and also exhibit dense contextual reasoning skills. To that end, we make use of a novel masking-based strategy that operates in the space of high-level assignment vectors that are produced in an online and standalone fashion (i.e., without the need for any pre-trained models or multiple training stages) using a teacher-student scheme and an online generated codebook. We have experimentally shown that our approach possesses significant computational advantages compared to prior methods, while exhibiting strong performance, not only in the case of full finetuning but also in the case of linear probing, k-NN, and low-shot settings. It thus allows us to obtain high-quality image representations with much smaller training computational budget compared to prior state-of-the-art approaches.

**Broader Impact Statement** Self-supervised image representation learning can exacerbate existing biases within pre-training data, potentially leading to biased visual recognition systems that disproportionately affect individuals based on gender, ethnicity, skin tone, or geographical location. Additionally, the collection of large-scale pre-training image datasets raises substantial privacy and copyright concerns, particularly when images contain sensitive, protected or personal information. For instance, unauthorized access or misuse of such data can result in privacy breaches, with potential repercussions for individuals depicted in the dataset. Therefore, practitioners deploying self-supervised pre-training in real-world applications must exercise caution, considering factors such as biases, privacy, and licensing associated with the pre-training data.

**Acknowledgements** This work was performed using HPC resources from GENCI-IDRIS (Grants 2022-AD011012884R1 and 2022-AD011013413), and from CINES under the allocation GDA2213 for the Grand Challenges AdAstra GPU made by GENCI. This work was supported by the Ministry of Education, Youth and Sports of the Czech Republic through the e-INFRA CZ (ID:90254) and by CTU Student Grant SGS21184OHK33T37. This research received the support of EXA4MIND, a European Union´s Horizon Europe Research and Innovation programme under grant agreement N° 101092944. Views and opinions expressed are however those of the author(s) only and do not necessarily reflect those of the European Union or the European Commission. Neither the European Union nor the granting authority can be held responsible for them.

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

# Appendix

## A  Additional experimental results

### A.1  Training without the $L_{\mathrm{IMG}}$ objective

In Tab. 8, we report more results from MOCA trainings with only the $L_{\mathrm{LOC}}$ loss and the $L_{\mathrm{IMG}}$ loss deactivated (i.e., setting $\lambda = 0$). The decoder of these models is implemented using: (a) the default approach described in Sec. 3.2, or (b) the condenser-based approach described in Sec. 3.3. We also provide results for when having two decoders (c), one using the default approach and the other using the condenser-based approach, each trained with a separate $L_{\mathrm{LOC}}$ loss.

We observe that, with only the $L_{\mathrm{LOC}}$ loss, the default decoder (model (a)) is better than the condenser-based decoder (model (b)), which is the opposite from the behavior when both the $L_{\mathrm{LOC}}$ and the $L_{\mathrm{IMG}}$ objectives are used. The fact that the optimal choice for the decoder is different depending on whether $L_{\mathrm{IMG}}$ is used or not hints that the two objectives work together in a synergistic way. Nevertheless, even with the default decoder (model (a)) the k-NN and Linear probing performances are still quite low. However, model (c), which combines the two decoders, fares much better. Indeed, in this case, the model better bootstraps its learning process and, thus, learns better image representations.

We also compare with variants where the *hide-and-predict* objectives are defined in the *pixel space* with per-patch normalization as in MAE (He et al., 2022) ('Pixels'). We see that in this pixel-reconstruction case, the decoder type has a small impact. Also, our assignment-prediction models (a) and (c) achieve significantly better k-NN and Linear probing performance than these pixel-reconstruction models, which shows the interest in exploiting higher-level features.

### A.2  Alternative codebook updating approach

During our pre-training, we update the codebook $C$ by randomly selecting teacher-token embeddings from previous mini-batches. However, in this exploration, we tested an alternative approach using farthest point sampling (FPS) to select teacher-token embeddings for the codebook updates. Following the pre-training setup outlined in Sec. 4.1, the use of FPS resulted in a decline in k-NN performance from 71.8% to 70.3%. We hypothesize that the reason for this drop in performance is that this FPS strategy is more prone to inserting outlier tokens to the codebook and/or the codebook changes more drastically from one iteration to the next, making the learning process harder.

### A.3  Longer pre-training

As we see in Table 9, longer pre-training in MOCA improves performance, but with diminishing returns. This behavior is similar to other pre-training methods, where performance converges after a certain number of pre-training epochs. MOCA's advantage lies in its faster convergence, effectively reducing pre-training time.

### A.4  Bigger backbones

As we see in Tables 9 and 10, MOCA can scale with bigger backbones and achieves strong results while it has smaller pre-training time.

## B  Creating the condenser-based decoder inputs

Here we describe in more detail how we produce the inputs for the condenser-based decoder of MOCA models. For each masked view $\tilde{\mathbf{x}}^v$, we create the decoder's input embeddings from the following student-produced embeddings:

Table 8: **Hide-and-predict models trained without the $L_{\text{IMG}}$ loss ($\lambda = 0$).** ImageNet-1k classification results with ViT-B/16 models trained for 100 pre-training epochs with a single token masking round. For these models, we use the default and/or the condenser decoder versions.

| | Decoder approach | | | Top-1 accuracy | |
|---|---|---|---|---|---|
| | Default | Condenser | Targets | k-NN | Linear |
| (a) | ✓ | | Assign. | 34.4 | 51.6 |
| (b) | | ✓ | Assign. | 13.1 | 33.7 |
| (c) | ✓ | ✓ | Assign. | **52.5** | **64.2** |
| (d) | ✓ | | Pixels | 14.6 | 38.8 |
| (e) | | ✓ | Pixels | 15.6 | 42.7 |
| (f) | ✓ | ✓ | Pixels | 13.9 | 41.9 |

Table 9: **Scaling pre-training epochs or model size.** Top-1 accuracy results on ImageNet with ViT-B/16 or ViT-L/16.

| Method | #Epoch | k-NN | Linear | Finetun. | 1-shot | 2-shot | 5-shot |
|---|---|---|---|---|---|---|---|
| `MOCA` ViT-B/16 | 200 | 77.2 | 78.7 | 83.6 | 55.4 | 63.9 | 69.9 |
| `MOCA` ViT-B/16 | 400 | 77.6 | 79.3 | 83.7 | 55.5 | 64.6 | 69.9 |
| `MOCA` ViT-L/16 | 200 | **78.4** | **80.3** | **84.9** | **56.3** | **65.2** | **70.7** |

Table 10: **ViT-L/16 comparisons.** Top-1 accuracy results on ImageNet. The time measurements are based on a single 8 A100 GPU node. †: use of multiple crops (Caron et al., 2021).

| Method | #Epoch | Time | k-NN | Linear | Finetun. |
|---|---|---|---|---|---|
| MAE | 1600 | 187h | - | 75.1 | **85.9** |
| iBOT† | 300 | 320h | 78.0 | **81.0** | 84.8 |
| `MOCA` | 200 | 89h | **78.4** | 80.3 | 84.9 |

1. The average patch-token embedding at the output of S (i.e., the last transformer layer of S): $\bar{\text{S}}(\tilde{\mathbf{x}}^v) = \frac{1}{|u_v|}\sum_{i \in u_v} \text{S}(\tilde{\mathbf{x}}^v)_i$. Note that in ViTs, after the last transformer layer, there is a `LayerNorm` layer. The patch-token embeddings $\{\text{S}(\tilde{\mathbf{x}}^v)_i\}_{i \in u_v}$ are the outputs of this `LayerNorm` layer.

2. The patch-token embeddings $\{\text{S}^\ell(\tilde{\mathbf{x}}^v)_i\}_{i \in u_v}$ from an intermediate transformer layer $\ell$ of S. As before, these intermediate-layer embeddings have been passed from an additional `LayerNorm` layer.

Before feeding these embeddings to the decoder, we first apply a linear projection layer $p(\cdot)$ in order to convert their dimensionality from $d_{\text{enc}}$ to $d_{\text{dec}}$. Then, we append a learnable masked embedding $\mathbf{e}^M$ for each masked token $i \in m_v$ and add the (non-learnable sin-cos-based) positional embeddings $\{\mathbf{e}_i^P\}_{i=0}^N$, where $\mathbf{e}_0^P$ is the positional embedding for the [AVG] token (i.e., for $\bar{\text{S}}(\tilde{\mathbf{x}}^v)$). This produces the following input set of tokens embeddings:

$$\mathbf{z}^v = \{p(\bar{\text{S}}(\tilde{\mathbf{x}}^v)) + \mathbf{e}_0^P\} \cup \{p(\text{S}^l(\tilde{\mathbf{x}}^v)_i) + \mathbf{e}_i^P\}_{i \in u_v} \cup \{\mathbf{e}^M + \mathbf{e}_i^P\}_{i \in m_v}. \tag{7}$$

In the case of partial decoding, the set $m_v$ in the above Eq. (7) is replaced by the subset $m_v' \subset m_v$.

## C Additional implementation details

### C.1 Weight generation modules $\text{G}^d(\cdot)$ and $\text{G}^b(\cdot)$

As described in the main paper (Sec. 3.2.1), for masked same-view token assignment prediction (see Eq. (2)) and the masked cross-view average assignment prediction (see Eq. (5)), we use the weight generation modules $\text{G}^d(\cdot)$ and $\text{G}^b(\cdot)$ respectively. Like Gidaris et al. (2021), we use MLP networks with the following configuration:

$$\texttt{L2Norm} \rightarrow \texttt{Linear(768,1536)} \rightarrow \texttt{BatchNorm} \rightarrow \texttt{ReLU} \rightarrow \texttt{Linear(1536,d)} \rightarrow \texttt{L2Norm},$$

Table 11: **MOCA 's settings.** We define the masking percentage as $100 \cdot \frac{|m_v|}{N}$ and the percentage of predicted tokens (i.e., partial decoding) as $100 \cdot \frac{|m'_v|}{N}$.

(a) **MOCA 's model setting**

| Hyperparameter | Value |
|---|---|
| Codebook - size $K$ | 4096 |
| Codebook - new words $K_{new}$ per training step | 4 |
| Masking - mask percentage for 1st round | 55% |
| Masking - mask percentage for 2nd round | 75% |
| Masking - the percentage of predicted tokens | 20% |
| Decoder - intermediate layer $\ell$ for input tokens | 8 |
| Decoder - depth | 2 |
| Decoder - embedding size $d_{\text{dec}}$ | 512 |
| Decoder - self-attention heads | 16 |
| Loss - weighting parameter $\lambda$ | 0.5 |

(b) **MOCA 's optimization setting**

| Hyperparameter | Value/Method |
|---|---|
| Teacher momentum $\alpha$ | $0.99 \rightarrow 1.0$ (cosine annealing) |
| optimizer | AdamW (Loshchilov & Hutter, 2019) |
| base learning rate | 1.5e-4 |
| weight decay | 0.05 |
| optimizer momentum | $\beta_1 = 0.9$, $\beta_2 = 0.999$ |
| batch size | 2048 |
| learning rate schedule | cosine decay Loshchilov & Hutter (2017) |
| warmup epochs | 30 |

Table 12: **ImageNet-1k linear classification.**

| Hyperparameter | Value/Method |
|---|---|
| optimizer | SGD |
| base learning rate | 0.04 |
| weight decay | 0.0 |
| optimizer momentum | 0.9 |
| batch size | 1024 |
| training epochs | 100 |
| learning rate schedule | cosine decay Loshchilov & Hutter (2017) |
| augmentation | RandomResizedCrop |

where $\mathbf{d}$ is $d_{\text{dec}}$=512 for $\mathrm{G}^d(\cdot)$ and $d_{\text{enc}}$=768 for $\mathrm{G}^b(\cdot)$. Also, these MLPs have a skip connection that begins after the input **L2Norm** unit and concludes before the output **L2Norm** unit. Given as input the codebook $C = [\mathbf{c}_1, \ldots, \mathbf{c}_K]$, $\mathrm{G}^d(\cdot)$ and $\mathrm{G}^b(\cdot)$ produce the prototypes weights $W^d = \mathrm{G}^d(C) = [\mathrm{G}^d(\mathbf{c}_1), \ldots, \mathrm{G}^d(\mathbf{c}_K)]$ and $W^b = \mathrm{G}^b(C) = [\mathrm{G}^b(\mathbf{c}_1), \ldots, \mathrm{G}^b(\mathbf{c}_K)]$ respectively. The $L_2$-normalization in the **L2Norm** layers is across the channels dimension and the normalization in the **BatchNorm** layers is across the $K$ codebook items, which is the batch dimension in these MLP networks.

## C.2 Pre-training MOCA

Here we share additional implementation details for the pre-training of our **MOCA** representations. In Tab. 11a we provide the implementation details for the ViT-B/16-based **MOCA** model that we used for producing the results of Sec. 4.4 in the main paper and in Tab. 11b the optimization setting for its training. For the ViT-B/16, we follow the standard ViT architecture (Dosovitskiy et al., 2021) and implement it with 12 transformer layers, 12 attention heads, and $d_{\text{enc}} = 768$ channels. We use sine-cosine positional embeddings (Vaswani et al., 2017) for both the encoder and the decoder inputs. Also, as in OBoW (Gidaris et al., 2021), at the teacher side, during the average pooling operation for computing the reduced assignment vectors, we ignore the token assignment vectors on the edge / border patch tokens (i.e., we ignore the 2 rows/columns of border patch tokens from each image side).

In order to generate the two unmasked random views $\mathbf{x}^1$ and $\mathbf{x}^2$, input for the teacher, we employed the widely used augmentation strategy found in most prior works, such as SimCLR (Chen et al., 2020a) and MoCo-v3 (Chen et al., 2021). The PyTorch pseudo-code for this augmentation stategy is provided in Appendix C.4. To generate the masked views that the student gets as input, we simply applied random token masking to the $\mathbf{x}^1$ and $\mathbf{x}^2$ random views.

Table 13: **Full fine-tuning for ImageNet-1k classification.**

| Hyperparameter | Value/Method |
|---|---:|
| optimizer | AdamW (Loshchilov & Hutter, 2019) |
| base learning rate | 1.0 -3 |
| weight decay | 0.05 |
| optimizer momentum | $\beta_1 = 0.9, \beta_2 = 0.999$ |
| layer-wise lr decay (Clark et al., 2020; Bao et al., 2022) | 0.65 |
| batch size | 1024 |
| learning rate schedule | cosine decay (Loshchilov & Hutter, 2017) |
| training epochs | 100 |
| warmup epochs | 5 |
| drop path (Huang et al., 2016) | 0.2 |
| label smoothing Szegedy et al. (2016) | 0.1 |
| mixup (Zhang et al., 2018) | 0.8 |
| cutmix (Yun et al., 2019) | 1.0 |
| augmentation | RandAug $(9, 0.5)$ (Cubuk et al., 2020) |

## C.3 Evaluation protocols

Here we report implementation details for the evaluation protocols that we used in our work. We note that in all of them we use the pre-trained teacher transformer.

**k-NN ImageNet classification.** For the k-NN evaluation protocol (Wu et al., 2018; Caron et al., 2021) we freeze the pre-trained transformer, extract features from the training images, and then for the test images use a k-nearest neighbor classifier with $k = 20$ neighbors.

**Linear ImageNet classification.** In this case, we again freeze the pre-trained transformer and train a linear classifier on the extracted training image features. We follow the linear classification setting of iBOT (Zhou et al., 2022). In Tab. 12 we provide implementation details and hyperparameter values.

**End-to-end fine-tuning on ImageNet classification.** We follow the end-to-end fine-tuning setting of iBOT (Zhou et al., 2022). In Tab. 13 we provide implementation details and hyperparameter values.

**Low-shot ImageNet classification.** Following MSN (Assran et al., 2022), we freeze the pre-trained transformer, extract image features from the few available training images, and then train a linear classifier with them using L2-regularized logistic regression. For the logistic regression, we use the **cyanure** package (Mairal, 2019).

**Cityscapes semantic segmentation with linear probing.** We study the quality of learned representations using linear probing either with the entire Cityscapes (Cordts et al., 2016) dataset containing 2975 training images or in the few-shot setup. For all experiments we evaluate the 500 validation images and report the best mean intersection over union (mIoU). We use the Segmenter (Strudel et al., 2021) model with a frozen ViT-B/16 backbone initialized with one of the studied approaches. Then, we append a single learnable linear layer on top.

*Entire dataset*: When training linear probes using the entire dataset, we optimize the network for 216 epochs following the protocol of Strudel et al. (2021). We optimize the learning rate for every studied method separately.

In the *few-shot setup*, we randomly sample three training subsets of the Cityscapes dataset that are shared across all the experiments with different methods. We use either splits of 100 or 374 training images (French et al., 2020). For every method, we search for the optimal learning rate on the first split and then use it for the remaining two training splits. We optimize every split for 100 epochs.

**Cityscapes semantic segmentation with fine-tuning.** Furthermore, we study different methods when used as an initialization of the Segmenter backbone when fine-tuning the entire network. In this setup, we

use a single layer of mask transformer (Strudel et al., 2021) as a decoder. As in the linear probe setup, we use either the entire dataset or 100-/372-large training splits and report the best mIoU on the validation set.

*Entire dataset*: We train all methods for 216 epochs as done by Strudel et al. (2021) and optimize the learning rate for each method separately.

*Few-shot setup*: We use the identical training splits as in the few-shot linear probing and search for the optimal learning rate in the same way. We optimize every method for 216 epochs.

### C.4   Image augmentations pseudo-code

Here we provide PyTorch pseudo-code for the image augmentations used in `MOCA` for generating the two unmasked random views $\mathbf{x}^1$ and $\mathbf{x}^2$.

```python
import torchvision.transforms as T
normalize = T.Normalize(mean=(0.485, 0.456, 0.406), std=(0.229, 0.224, 0.225))

aug_view1 = T.Compose([
    T.RandomResizedCrop(224, scale=(0.2, 1.)),
    T.RandomApply([T.ColorJitter(0.4, 0.4, 0.2, 0.1)], p=0.8),
    T.RandomGrayscale(p=0.2),
    GaussianBlur(1.0, 0.1, 2.0),
    T.RandomHorizontalFlip(),
    T.ToTensor(),
    normalize])

aug_view2 = T.Compose([
    T.RandomResizedCrop(224, scale=(0.2, 1.)),
    T.RandomApply([T.ColorJitter(0.4, 0.4, 0.2, 0.1)], p=0.8),
    T.RandomGrayscale(p=0.2),
    GaussianBlur(0.1, 0.1, 2.0),
    Solarization(0.2),
    T.RandomHorizontalFlip(),
    T.ToTensor(),
    normalize])
```

