# OpenReview forum: "MOCA: Self-supervised Representation Learning by Predicting Masked Online Codebook Assignments"
_TMLR — Accepted by TMLR_

### Review · Reviewer_fwxy · 2023-10-02

**Summary Of Contributions:**

The paper proposes self-supervised learning techniques based on mask-and-predict and the codebook models. The idea is to mask the image randomly and then aggregate the feature representation using codebooks, and then use the masked embedding to predict the non-masked codebook representation (under augmentations). The idea is interesting and sort of combining new methods with old techniques. Codebook methods provide some level of perturbation invariance to the learning procedure. The paper performed experiments and demonstrated the effectiveness of the proposed method (MOCA). The results are impressive.

In general, I think the paper has novelty and shows promise, with some concerns on the complication. Overall positive.

**Audience:**

Yes

**Broader Impact Concerns:**

The authors have stated the concerns clearly in the paper which I tend to agree with. There will be bias and copyright issues coming up with these SSL approaches.

**Claims And Evidence:**

Yes

**Requested Changes:**

I think the introduction of the methods can be improved by highlighting the following questions:

1. How does the codebook construction look like? There are teachers and students. Do they share the same codebook? If the system is cold-started, then how does the codebook look like in the beginning? I suggest organize a separate section just about codebook construction.

2. Codebook methods are usually resulting into a histogram but I don't see the proposed method goes that way. Do you only quantize the feature representations so that it becomes more stable?

3. How does global average pooling contribute to the final performance?

4. It is surprising to me the optimal lambda is 0.5, can you elaborate on this? Have you evaluated it on multiple datasets and how would its optimal value change across different datasets?

**Strengths And Weaknesses:**

Strength

1. The proposed idea is very interesting. Codebook models were very popular ten years ago in computer vision. The authors combine the idea of codebooks into self-supervised learning which I think is very smart and novel.
2. The results are impressive. The proposed method basically outperform existing SOTA methods in SSL. This demonstrates the authors' hypothesis that a moderate level of perturbation invariance in feature representation helps with self supervised learning.
3. The paper is nicely written with sufficient experimental validation and ablation study.

Weakness

1. Although I feel the paper is generally well written, the method intro can be a bit more clear. Some key procedures for codebook construction and assignments are not emphasized or highlighted, which I believe are core questions for codebook methods.
2. Although the approach combines codebook idea and masking, related to weakness 1, I generally feel the proposed method is a bit complicated. Without sufficient differences, I am worried the proposed method may be not adopted by practitioners due to its addition complication.

---

> ### Author Response · Authors · 2023-12-15
> **Response to Reviewer fwxy - Part 1/2**
>
> > About the codebook
>
> **Answer.** We thank the reviewer for pointing out this lack of clarity in our methodology description. To address these concerns, we provide detailed responses:
>
> - The codebook $C$ is essentially constructed as a K-sized queue, following the methodology of OBoW [Gidaris et al., 2021]. At each training step, $K_{\text{new}}$ teacher patch-token embeddings are randomly selected from the current mini-batch, one from each randomly selected image in the mini-batch. In particular, after randomly selecting with uniform distribution $K_{\text{new}}$ images from the mini-batch, then we proceed by randomly selecting (again with uniform distribution) one teacher patch-token embedding from each image. These embeddings replace the oldest $K_{\text{new}}$ embeddings in the queue. This dynamic process forms the codebook $C$ for producing code assignments at the teacher's side.
>
> - At the very beginning (training step 0), the codebook embeddings in $C$ are randomly initialized from a normal distribution (zero mean, unit standard deviation). Since, the learning rate at the first training steps (during which the queue still has randomly initialized embeddings) is very low due to the warmup period, this random initialization does not affect the learning process.
>
> - For the student-side prediction tasks (masked same-view token assignment and masked cross-view average assignment), we employ MLP-based weight generation modules to dynamically generate the weight matrices $W^d$ and $W^b$ used in equations (2) and (5) respectively. This process is described in Sec. 3.5 and the appendix Sec. C.1. Briefly, these MLP-based weight generation modules take as input the codebook $C$ and produce a weight vector for each embedding in $C$. The dynamically generated weight matrices $W^d$ and $W^b$ can be seen as the codebooks at the student side.
>
> To enhance clarity:
> - In Sec. 3.1, we will elaborate on codebook construction and initialization when discussing the extraction of teacher-produced token assignment vectors.
> - A dedicated sub-subsection, 3.2.1, will be added to cover the dynamic weight generation modules, incorporating information from Sec. 3.5 and appendix Sec. C.1.
>
> We believe these additions will significantly improve the clarity of the codebook construction and usage. Does the reviewer find these changes satisfactory?
>
> [Gidaris et al., 2021] Spyros Gidaris, Andrei Bursuc, Gilles Puy, Nikos Komodakis, Matthieu Cord, and Patrick Perez. OBoW: Online bag-of-visual-words generation for self-supervised learning. In CVPR, 2021.
>
> > Proposed method is a bit complicated
>
> **Answer.** In addressing concerns about complexity, it's crucial to note that MOCA eliminates the need for additional pre-training stages like VQ-VAE for image tokenization—a distinctive simplicity compared to some other codebook-based masking methods. Furthermore, our online codebook construction method is actually quite simple, utilizing a queue of randomly selected patch-token embeddings.
>
> While it's true that teacher-student methods, such as DINO, iBOT, MSN and our MOCA method, may appear more complicated in design (due to the challenge of learning good representations with randomly initialized teacher and student, while avoiding collapse), they outperform simpler pixel reconstruction approaches by learning highly effective "ready-to-use" representations.
>
> Notably, MOCA achieves superior results in low-shot settings while using a significantly smaller pre-training budget, making it a good choice for practitioners with limited computational resources.
>
> > Codebook methods result into a histogram
>
> **Answer.** As explained in the “Masked cross-view average assignment prediction” paragraph at the end of Sec. 3.2,  we reduce the teacher assignment vectors to a single K-dimensional ‘bag-of-words’ (BoW) representation using global average pooling. This BoW representation can be seen as a histogram where each channel encodes the frequency / likelihood with which each codebook item appears in the image.
>
> > Global average pooling contribution
>
> **Answer:** We evaluate the impact of using global average pooling in Tab. 2(d) and observe that we obtain significantly better results when using an average pooling then using the [CLS] token. We hypothesize that the [AVG] token leads to learning assignment vectors that encode more high-level concepts. We have a discussion in Sec. 4.1.

---

> > ### Author Response · Authors · 2023-12-15
> > **Response to Reviewer fwxy - Part 2/2**
> >
> > > About optimal lambda.
> >
> > **Answer.** We use a coarse grid (consisting of 0, 0.25, 0.50, 0.75, and 1.0) for searching the optimal lambda. From the results in Tab. 2(e) we see that the best two lambda values are 0.5 (`71.8%` $k$-NN accuracy) and 0.25 (`71.5%` $k$-NN accuracy), with small performance difference between them. This means that the performance of MOCA is relatively stable between lambda values 0.25 and 0.50, with perhaps an even better lambda choice lying in this interval (or somewhere near it). Overall this shows that the local patch-wise objective $L_{\mathrm{LOC}}$ should have similar strength (i.e., weight) as the image-wise objective $L_{\mathrm{IMG}}$.
> >
> > Exhaustively evaluating this choice on several downstream datasets would be impractical. But, indeed it could be the case that the optimal lambda for another downstream task/dataset could be different. For example, a dense prediction downstream tasks might require a smaller lambda (e.g., closer to 0.25), that puts more emphasis on the local patch-wise predictions.

---

### Review · Reviewer_hvsU · 2023-10-15

**Summary Of Contributions:**

This paper introduced a self-supervised learning approach that unifies masking-and-prediction and invariance to perturbation paradigms. In particular, a teacher-student framework is introduced and the student is forced to mimic the predictions made by the teacher network on unlabeled data for representation learning. Losses are defined at two levels where both global and local (dense) representations are aligned. Experiments on Knn classification, linear probing and fine-tuning on two datasets, including ImageNet-1k and CityScapes, suggest the superiority of the proposed method.

**Audience:**

Yes

**Broader Impact Concerns:**

There is no concern over ethical implications.

**Claims And Evidence:**

No

**Requested Changes:**

1. It is necessary to present details of data augmentation applied. It is also highly recommended to evaluate how different augmentation strategy could affect the results.

2. The specific design of memory bank update needs to be further elaborated. I suggest evaluating update the memory bank with farthest sampling strategy.

3. More diverse downstream tasks are for sure necessary. These could include more segmentation tasks, e.g. Pascal VOC, MS COCO, etc. as dense contextual reasoning is claimed to be the novelty.

**Strengths And Weaknesses:**

Strength:

1. Self-supervised pre-training is a very important task in representation learning. An improved self-supervised learning method can potentially contribute to many downstream tasks.

Weakness:

1. It is not clear what data augmentation is applied for student and teacher views respectively.

2. The specific design of updating the memory bank storing teacher-token deserves further investigation. Randomly sampling token embeddings to replace old tokens in the memory is sensible to non-i.i.d. sampling of training samples. A more stable way of updating the memory bank might be using farthest sampling to select most diverse tokens to update the memory bank.

3. The fine-tuning performance between different self-supervised learning approaches are rather close according to Tab. 4 (a). Therefore, it begs the question whether the more complicated self-supervised pre-training is really necessary.

4. Downstream tasks are only evaluated on two datasets, i.e. ImageNet classification and CityScapes segmentation. The evaluation is far from enough for the purpose of extensively evaluating the effectiveness of a new self-supervised pre-training method.

---

> ### Author Response · Authors · 2023-12-15
> **Response to Reviewer hvsU**
>
> > It is necessary to present details of data augmentation applied. It is also highly recommended to evaluate how different augmentation strategies could affect the results.
>
> **Answer.** In order to generate the two unmasked random views x1 and x2, input for the teacher, we employed the widely used default augmentation strategy found in most prior works such as SimCLR, DINO, MOCO-v3, iBOT, etc. In particular, the pytorch (pseudo-)code for this augmentation strategy is:
>
> ```python=
> import torchvision.transforms as T
> normalize = T.Normalize(mean=(0.485, 0.456, 0.406), std=(0.229, 0.224, 0.225))
>
> aug_view1 = T.Compose([
>     T.RandomResizedCrop(224, scale=(0.2, 1.)),
>     T.RandomApply([T.ColorJitter(0.4, 0.4, 0.2, 0.1)], p=0.8),
>     T.RandomGrayscale(p=0.2),
>     GaussianBlur(1.0, 0.1, 2.0),
>     T.RandomHorizontalFlip(),
>     T.ToTensor(),
>     normalize])
>
> aug_view2 = T.Compose([
>     T.RandomResizedCrop(224, scale=(0.2, 1.)),
>     T.RandomApply([T.ColorJitter(0.4, 0.4, 0.2, 0.1)], p=0.8),
>     T.RandomGrayscale(p=0.2),
>     GaussianBlur(0.1, 0.1, 2.0),
>     Solarization(0.2),
>     T.RandomHorizontalFlip(),
>     T.ToTensor(),
>     normalize])
> ```
>
> We will provide these details in the revised version of our manuscript. To generate the masked views that the student gets as input, we simply applied random token masking to the x1 and x2 random views.
>
> We believe that ablating the default augmentation strategy goes beyond the scope of our work. Instead, our primary focus lies in ablating the percentage of token masking applied to the student’s input, as outlined in Tab. 2c, which is the main difference of our overall augmentation strategy with those employed in prior works.
>
> > The specific design of memory bank update needs to be further elaborated. I suggest evaluating update the memory bank with farthest sampling strategy.
>
> **Answer.** We thank the reviewer for the suggestion of using the farthest point sampling (FPS) strategy during the updating of the codebook. We tried this experiment and we did not see any improvement in performance. Specifically, using FPS dropped the k-NN performance from `71.8%` to `70.3%`. We hypothesize that the reason for this drop in performance is that this FPS strategy is more prone to inserting outlier tokens to the codebook and/or the codebook changes more drastically from one iteration to the next, making the learning process harder. We will add these results in the revised version of our manuscript.
>
> > The fine-tuning performance between different self-supervised learning approaches are rather close according to Tab. 4 (a). Therefore, it begs the question whether the more complicated self-supervised pre-training is really necessary.
>
> **Answer.** Certainly, ImageNet fine-tuning proves to be a nearly-saturated downstream task, with all pre-training methods showing similar performance. This leads us to explore low-shot settings, such as ImageNet classification and Cityscapes semantic segmentation, which we find more challenging and, in our opinion, of high practical interest. In these settings, MOCA showcases notable performance improvements. Additionally,  MOCA offers a crucial advantage—it achieves these gains while utilizing significantly smaller pre-training computation budgets.
>
> > Downstream tasks are only evaluated on two datasets, i.e. ImageNet classification and CityScapes segmentation. The evaluation is far from enough for the purpose of extensively evaluating the effectiveness of a new self-supervised pre-training method. More diverse downstream tasks are for sure necessary. These could include more segmentation tasks, e.g. Pascal VOC, MS COCO, etc. as dense contextual reasoning is claimed to be the novelty.
>
> **Answer.** We'd like to highlight that we have already assessed our method on MS COCO, specifically through the object detection and instance segmentation tasks. The corresponding results are provided in the appendix Sec. A.4. In the revised version of our manuscript, we will relocate this section to the main paper for better visibility.

---

### Review · Reviewer_JVWW · 2023-10-19

**Summary Of Contributions:**

This work deals with the challenge of learning self-supervised visual representation. It proposes to unify the discriminative and generative pretraining frameworks so the learned features are both semantically meaningful and contextual reasoning. Experimental results show that the proposed method is more data-efficient than previous methods.

**Audience:**

Yes

**Broader Impact Concerns:**

I have no concerns about the ethical implications of the work.

**Claims And Evidence:**

Yes

**Requested Changes:**

Please see the weaknesses stated above.

**Strengths And Weaknesses:**

The paper proposed a novel framework that combines discriminative and generative pretraining so the advantage from both sides can be gained, i.e., the learned features are both ready-to-use (leading to high kNN and linear acc) and detailed (leading to high finetuned acc) features. As a result, the training efficiency and data efficiency are improved.

My concerns include:

1. I find that the proposed method is more data-efficient than existing methods very interesting. However, I find little insight to help understand why it performs particularly better in the low-shot settings but is not so good in the data-sufficient settings.

2. From my point of view on self-supervised pretraining, the evaluation protocol of finetuning (including low-shot) and transfer learning is more important than kNN and linear probing, as they are more practical and show the value of pretraining. However, over half of this paper’s comparisons and conclusions are based on kNN and linear accuracies. For example, in Table 3, the model performs better without the condenser in finetuning. Plus, I hope to see more ablation studies on semantic segmentation experiments to learn the effect of each component of the framework. It is very likely that I am biased, and I am eager to hear from the authors about why they believe kNN and linear accuracies are so important.

3. Some discussions may be questionable. For example, in Section 4.1, the authors say “On the other hand, MAE’s decoder, which must reconstruct pixels, is better to be deeper and thus spare the MAE’s encoder from capturing such low-level details.” However, in SimMIM, which also reconstructs pixels, only one linear layer is used in the decoder.

---

> ### Author Response · Authors · 2023-12-15
> **Response to Reviewer JVWW - Part 1 / 2**
>
> > Why it is better in data-efficient settings?
>
> **Answer.** Indeed, our method exhibits superior performance improvements in low-shot scenarios, such as low-shot ImageNet classification and (low-shot) Cityscapes semantic segmentation, compared to data-sufficient settings like linear probing or fine-tuning with the entire ImageNet dataset. We believe that one of the main reasons for its notable success in data-efficient settings is our condenser decoder design. This design promotes the student's global image representation to capture more spatial structure from the input view—an aspect we deem crucial for data-efficient scenarios. On top of that, our method combines objectives that enforce both invariance to perturbations (image-wise loss) and dense feature generation (local patch-wise loss).
>
> Additionally, even in data-sufficient settings, our method achieves results on par with the state-of-the-art, despite utilizing a significantly smaller pre-training budget. It's worth noting that in these data-sufficient settings, there may be limited room for performance improvement as the amount of training data and training iterations can compensate for potential shortcomings of the original pre-trained representations or weights. For example, the differences in pre-training methods on ImageNet fine-tuning settings are marginal.
>
> > About the evaluation protocol
>
> **Answer.** We appreciate the reviewer’s thoughtful perspective on the evaluation protocol. We note that we do present results on various tasks, such as ImageNet classification, Cityscapes semantic segmentation, and MS COCO object detection (detailed in the appendix Sec. A.4; we will relocate this section to the main paper for better visibility). Also, although we understand the emphasis on the practicality of fine-tuning, we note that in extreme low-shot scenarios, like one training example per class, fine-tuning becomes less practical due to the risk of overfitting, and the performance improvement is uncertain.
>
> Pre-training strategies aiming for good initialization for fine-tuning remain of particular interest when both the amount of unlabeled pre-training data and the downstream labeled data are abundant and the computational budget is not constrained.
>
> Our focus on $k$-NN and linear probing stems from the belief that self-supervised methods should yield "ready-to-use" representations that can be used without requiring extensive fine-tuning. In our view, the ultimate goal of self-supervised pre-training is to provide such general-purpose representations that excel in out-of-the-box scenarios, similar to how successful models like GPT are used in NLP without fine-tuning.
>
> Also, we argue that $k$-NN and linear probing better indicates the network's ability to extract meaningful representations compared to using pre-trained networks solely as initialization for fine-tuning. For instance, ImageNet fine-tuning is a nearly-saturated downstream task, where, as evidenced in Tab. 9 of the iBOT paper, even models that collapsed during training can exhibit strong fine-tuning performance.
>
> In conclusion, $k$-NN and linear probing serve as more robust downstream tasks for evaluating the quality of learned representations, aligning with our vision of what self-supervised methods should strive to achieve. We hope this clarification addresses the reviewer’s concerns and provides insight into our evaluation protocol.
>
> > More ablation studies on semantic segmentation experiments
>
> **Answer.** We ablate the use of the local patch-wise loss $L_{\mathrm{LOC}}$, the condenser decoder design, and 2nd masking round on the low-shot Cityscapes semantic semgentation with linear probing, using 100 or 374 training images. The mIoU results are:
> | Objectives | Condenser | #Masks | mIoU 100 train imgs | mIoU 374 train imgs |
> |:----------:|:---------:|:------:|:--------------:|:--------------:|
> | $L_{\mathrm{IMG}}$ | N/A | 1 |      44.6      |      50.9      |
> | $L_{\mathrm{IMG}}$ & $L_{\mathrm{LOC}}$ | no | 1 |      44.2      |      51.5      |
> | $L_{\mathrm{IMG}}$ & $L_{\mathrm{LOC}}$ | yes| 1 |      46.8      |      52.4      |
> | $L_{\mathrm{IMG}}$ & $L_{\mathrm{LOC}}$ | yes| 2 |      49.9      |      55.2      |
>
> We will add these results in the revised version of manuscript.

---

> > ### Author Response · Authors · 2023-12-15
> > **Response to Reviewer JVWW - Part 2 / 2**
> >
> > > Questionable discussion about MAE. In SimMIM, which also reconstructs pixels, only one linear layer is used in the decoder.
> >
> > **Answer.** We note that a similar observation about the MAE's decoder is also present in the MAE paper itself. We cite from the MAE paper verbatim:  *“A sufficiently deep decoder is important for linear probing. This can be explained by the gap between a pixel reconstruction task and a recognition task: the last several layers in an autoencoder are more specialized for reconstruction, but are less relevant for recognition. A reasonably deep decoder can account for the reconstruction specialization, leaving the latent representations at a more abstract level.”*
> >
> > Regarding SiMIM, it is crucial to highlight a key difference between the SiMIM architecture and the architectures employed in MAE and our MOCA work. MAE and MOCA use encoder-decoder architectures. By that we mean that the encoder only processes the visible / unmasked tokens and it is only the decoder that processes the masked tokens. In contrast, SiMIM is essentially an encoder-only architecture. By that we mean that the encoder processes both the unmasked and the masked tokens. So, in fact, to reconstruct pixels the SiMIM architecture uses both the encoder, which typically has more transformer layers than the decoder in MAE, as well as the final linear prediction layer.

---

### Review · Reviewer_J994 · 2023-10-20

**Summary Of Contributions:**

The paper proposes a self-supervised pre-training strategy for ViTs. The approach involves using a masking and predict strategy where the teacher produces code book assignments from unmasked views and the student is trained to predict them using masked views. Additionally, an image-wise loss on averaged features is also added to enforce global consistency hoping to learn good linearly separable features.

Authors show that the approach sees a fast convergence which leading to better results on selected benchmarks.

**Audience:**

Yes

**Broader Impact Concerns:**

The authors have provided a broader impact section - don't have any additional concerns here.

**Claims And Evidence:**

Yes

**Requested Changes:**

Please refer to the weakness section for comments. Apart from some clarity fixes which might be easy to do, a more extensive transfer learning evaluation would help strengthen the paper.

**Strengths And Weaknesses:**

Strengths:
- Self-supervision is expensive to train. Being able to reduce training time while showing better performance is a good direction and helpful to the community.
- The proposed approach picks up some interesting bits from prior works (SwAV, DINO, MAE..) with some novel ideas and combines them in a unified framework.
- Overall, the paper is well written. Experiments and ablations are well thought of and executed.
- Strategy for partial decoding and multiple masking rounds is quite interesting.
- Promising results and comparison with prior works in this area.

Weaknesses/Questions:
- Unclear why student and teachers work with different 'prototypes'. More intuitions and details on this would be helpful. A short paragraph on how this relates to an approach like SwAV would also help with the motivation.
- Currently the global loss operates on an augmented version of the image and a masked version of the image. While masking can be considered to be an augmentation, it is a lot more extreme. Did you also try to add the IMG loss on unmasked student inputs (like how discriminative methods did in the past) ?
- More intuitions (and perhaps a figure) needed for the 'condenser' module since it significantly affects the performance.
- I think the paper needs more transfer learning and few-shot learning experiments. You can perhaps use a benchmark like this [a] or a more recent benchmark to compare discriminative and MIM approaches.
- Missing comparisons on MaskFeat ?

[a] How Well Do Self-Supervised Models Transfer?

---

> ### Author Response · Authors · 2023-12-15
> **Response to Reviewer J994**
>
> > Unclear why student and teachers work with different 'prototypes'. How this relates to an approach like SwAV?
>
> **Answer.** Indeed, the prototypes employed by the student for predictions are dynamically computed from the weight generation modules using the teacher prototypes (codebook $C$) as input. We use this dynamic computation due to MOCA's ''asymmetric'' architecture. By that we mean on the teacher side the assignment vectors are produced from the patch-token embeddings of the teacher encoder while on the student the average assignment vector (image-wise objective $L_{\mathrm{IMG}}$) is predicted from the global image embedding of the student encoder and the patch-wise assignment vectors (local patch-wise objective $L_{\mathrm{LOC}}$) are predicted from the student decoder, whose embeddings have different dimensionality than those of the encoder.
>
> The distinct ways in which the student and the teacher utilize prototypes for predictions and target generation, respectively, necessitate the use of different prototypes. We have considered an alternative approach, which consists in employing projector layers on the global image embedding of the student encoder and the patch-wise embeddings for the student decoder. However, preliminary experiments yielded poor results.
>
> In contrast, SwAV adopts a more ''symmetric'' architecture, with no separate teacher network. Prototypes are used nearly identically for both target generation ("teacher" side) and predictions ("student" side). Consequently, SwAV utilizes the same prototypes on both sides, learned through SGD. Furthermore, SwaV uses only global features to match with the prototypes, while we make use of both local and global representations.
>
> > Did you also try to add the IMG loss on unmasked student inputs?
>
> **Answer.** Following the reviewer's suggestion, we attempted the suggested experiment. However, the model collapsed during training, yielding unsatisfactory results. Perhaps the removal of the masks impacts significantly the difficulty of the pretext task. In a teacher-student approach this can easily lead to collapse if the learning rate and/or the teacher's momentum coefficient are not properly re-adjusted. Unfortunately, due to time constraints, we were unable to perform this required re-adjustments.
>
> > More intuitions needed for the 'condenser' module
>
> **Answer.** The condenser aims at forcing the encoder's global image embedding to capture crucial spatial structure information necessary for the local token assignment prediction task. Indeed, the condenser design is based on the idea of creating a bottleneck in the local token assignment prediction task. This bottleneck is specifically formed by using patch-token embeddings from a lower transformer layer as input for the decoder. To compensate for this bottleneck, the decoder becomes more dependent on the global image embedding that it also receives as input.
>
> > I think the paper needs more transfer learning and few-shot learning experiments. You can perhaps use a benchmark like this [a] or a more recent benchmark to compare discriminative and MIM approaches.
> [a] How Well Do Self-Supervised Models Transfer?
>
> **Answer.** We appreciate the reviewer's suggestion regarding benchmark settings. Unfortunately, due to time constraints, we were unable to conduct the suggested evaluation experiments in our study. Nevertheless, it's worth noting that we have already assessed our method across three datasets (ImageNet, Cityscapes, and MS COCO) and under various transfer learning and few-shot learning tasks. These tasks include $k$-NN, linear probing, fine-tuning, and low-shot experiments on ImageNet classification. Additionally, we conducted fine-tuning and linear probing on Cityscapes for semantic segmentation, considering both the full training set and low-shot training splits. Finally, we assessed object detection and instance segmentation on MS COCO (detailed in the appendix Sec. A.4; we will relocate this section to the main paper for better visibility).
>
> > Missing comparisons on MaskFeat ?
>
> **Answer.** We appreciate the reviewer for highlighting this omission, and we will incorporate this work in the revised version of our manuscript. In MaskFeat, HOG features are used as reconstruction targets. While HOG features offer higher-level reconstruction targets compared to image pixels, they still encode more low-level information than the code assignment vectors in our approach.
>
> In terms of performance, MaskFeat achieves an `84.0%` accuracy in ImageNet fine-tuning, slightly outperforming our MOCA's `83.6%`. However, it exhibits significantly poorer linear probing performance. According to the MaskFeat paper (Tab. 17), MaskFeat's best linear probing accuracy is `67.7%` with ViT-L. In contrast, our approach achieves `78.7%` with ViT-B and `80.3%` with ViT-L, showcasing superior results in linear probing.

---

> > ### Comment · Reviewer_J994 · 2023-12-28
> > **Thanks for the rebuttal**
> >
> > Thanks for the rebuttal and the new ablations. This addresses most of my concerns.
> > A few final comments:
> > - I do not see changes to the paper. I urge the authors to make the proposed changes.
> > - "More intuitions needed for the 'condenser' module" - I suggest the authors include a figure as mentioned in my original review. - Given the extended time given to the rebuttal, I did expect additional transfer learning experiments. These would help strengthen the paper :
> >     - In my experience, the benchmark I pointed to is pretty quick to train; I have done single GPU experiments on many of those datasets. There might be newer versions of such a benchmark which could be used as well.
> >     - These experiments help cement the claim on "out-of-the-box" useful features as mentioned by the authors especially given differences in Table 9 are marginal (I do acknowledge that MOCA is trained on far fewer epochs)

---

### Decision · Action_Editor_hfVG · 2024-01-05

**Recommendation:** Accept as is

**Comment:**

Overall there seems to be agreement that this is a useful addition to the self-supervised learning literature, thanks to a few novel ideas as well as novel combination of previous ideas, that together lead to more efficient training of vision transformers.  Execution is also praised -- writing, ablation study -- and promising results.

**Audience:**

Self-supervised learning and its efficiency continue to be a topic of wide interest.

**Claims And Evidence:**

3 out of 4 reviewers were positive about the evidence in the paper after rebuttal. The fourth asked for results that the authors pointed out were in the appendix, in rebuttal.